# Heterogeneous A40926 Self-Resistance Profile in *Nonomuraea gerenzanensis* Population Informs Strain Improvement

Elisa Binda [1], Francesca Berini [1], Flavia Marinelli [1], Adriana Bava [2] and Fabrizio Beltrametti [2,*]

[1] Department of Biotechnology and Life Sciences, University of Insubria, Via J.H. Dunant 3, 21100 Varese, Italy; elisa.binda@uninsubria.it (E.B.); f.berini@uninsubria.it (F.B.); flavia.marinelli@uninsubria.it (F.M.)
[2] BioC-CheM Solutions Srl, Via R. Lepetit 34, 21040 Gerenzano, Italy; abava@bioc-chemsolutions.com
* Correspondence: fbeltrametti@bioc-chemsolutions.com

**Abstract:** *Nonomuraea gerenzanensis* ATCC 39727 produces the glycopeptide antibiotic A40926, which is the natural precursor of the semi-synthetic, last-resort drug dalbavancin. To reduce the cost of dalbavancin production, it is mandatory to improve the productivity of the producing strain. Here, we report that the exposure of *N. gerenzanensis* wild-type population to sub-inhibitory concentrations of A40926 led to the isolation of differently resistant phenotypes to which a diverse A40926 productivity was associated. The most resistant population (G, grand colonies) represented at least the 20% of the colonies growing on 2 μg/mL of A40926. It showed a stable phenotype after sub-culturing and a homogeneous profile of self-resistance to A40926 in population analysis profile (PAP) experiments. The less resistant population (P, petit) was represented by slow-growing colonies to which a lower A40926 productivity was associated. At bioreactor scale, the G variant produced twice more than the wild-type (ca. 400 mg/L A40926 versus less than 200 mg/L, respectively), paving the way for a rational strain improvement based on the selection of increasingly self-resistant colonies.

**Keywords:** dalbavancin; A40926; *Nonomuraea gerenzanensis*; glycopeptide antibiotics; antibiotic resistance; strain improvement

## 1. Introduction

*Nonomuraea gerenzanensis* ATCC 39727 is a filamentous actinomycete producing the glycopeptide antibiotic (GPA) A40926 [1], which is the natural precursor of the semi-synthetic derivative dalbavancin [2]. Dalbavancin is a second-generation GPA, which is in clinical practice to treat severe infections caused by multidrug-resistant, gram-positive pathogens [3]. Dalbavancin is today marketed in Europe and USA under the trade names Xydalba and Dalvance, respectively, and it has been the first antibiotic designated as a Qualified Infection Diseases Product by FDA because of its potency, extended dosing interval, and unique dose regimen (once a week). Although many efforts have been devoted in the last two decades to the improvement of its production process [4], its cost still largely exceeds that of first-generation GPAs, i.e., vancomycin and teicoplanin. Therefore, following the sequencing of the A40926 biosynthetic gene cluster (BGC), named *dbv*, in 2003 [5], multiple aspects of A40926 biosynthesis and its regulation were investigated with the aim to improve its production [6–8]. It was recently clarified that *N. gerenzanensis* produces the GPA in the form of *O*-acetyl-A40926 (with an *O*-acetylated mannose residue), but the acetyl group is lost during the alkaline extraction of the antibiotic [9,10]. Since it was this deacetylated GPA that was initially named A40926, we will continue to refer to it as A40926 hereafter. Self-resistance appeared co-regulated with A40926 antibiotic production, and it increased during the antibiotic biosynthesis [11,12]. Self-resistance was induced by the A40926 (the available antibiotic) but apparently repressed by its acetyl-form (the real end product of the biosynthesis), although through a regulatory circuit that it is still not

completely understood [10,13]. In this paper, we focused our attention on *N. gerenzanensis* wild-type population, which interestingly shows a heterogeneous self-resistance profile to the antibiotic A40926. Exposing *N. gerenzanensis* to sub-inhibitory concentrations of A40926 led to the isolation of differently resistant phenotypes to which a diverse A40926 productivity was associated, paving the way for a rational strain improvement based on the selection of increasingly resistant colonies.

## 2. Materials and Methods

### 2.1. Strains and Cultivation Conditions

*Nonomuraea gerenzanensis* ATCC 39727 was maintained as a lyophilized master cell bank (MCB). A working cell bank (WCB) was prepared from a first-generation slant originating from the MCB as previously described [14,15]. Cryo-vials of 1.5 mL were stored at –80 °C for up to six months without influence on the A40926 production during fermentation. Mycelium for plating and for the isolation of colonies was prepared as follows. Cryo-vials of the WCB were thawed at room temperature, and 2 mL were used to inoculate 100 mL of SM medium [16]. Strains were grown to the exponential phase (approximately 72 h) at 28 °C with shaking. Mycelium was then harvested by centrifugation, resuspended in 0.9% (*w/v*) NaCl (Merck KGaA, Darmstadt, Germany), and fragmented by sonication with Vibracell Albra sonicator 400 W model (Sonics and Materials Inc., Newtown, CT, USA) at an intensity of 6 kHz, pulse on, 5 s, and pulse off, 3 s, for a total interval sufficient to give single unbranched hyphae with size ranging from 1 to 5 μm (checked with microscope), as previously described [11–13]. The resulting cell suspension was filtered through a 5 μm Durapore membrane filter (Merck Millipore, Burlington, MA, USA), harvested by centrifugation, resuspended in fresh SM medium, and incubated for 3 h with shaking to revitalize the mycelium. The mycelium was either stored in 1 mL aliquots at –80 °C or immediately processed.

Fermentation for A40926 production was performed as follows. Cryo-vials of the WCB were thawed at room temperature, and 2 mL were used to inoculate 100 mL of E25 vegetative medium in 500 mL baffled flasks and grown for 72–96 h on a rotary shaker at 200 rpm and 28 °C. E25 medium components were in g/L: yeast autolysate (Costantino & C SpA, Favria, Italy) 4; soy bean meal (Cargill Srl, Wayzata, MN, USA) 20; dextrose (Cerestar Italia SpA, Castelmassa, Italy) 25; NaCl (Carlo Erba Reagents Srl, Cornaredo, Italy) 1.25; $CaCO_3$ (Baslini Spa, Milan, Italy) 5; and antifoam (Hodag; Vantage, Deerfield, IL, USA) 0.3; pH of the medium prior to autoclaving was adjusted to 7.5 with NaOH (Merck KGaA, Darmstadt, Germany) [11,12]. Fermentation was started by adding a 10% (vol/vol) inoculum from the vegetative medium flask into the FM2 production medium in 500 mL baffled flasks or in a 2 L working volume P-100 Applikon glass reactor (height, 25 cm; diameter, 13 cm; Applikon Biotechnology, Delft, The Netherlands) equipped with an AD1030 Biocontroller and AD1032 motor. FM2 components were in g/L: bacto-yeast extract (Costantino & C SpA, Favria, Italy) 8; soybean flour (Merck KGaA, Darmstadt, Germany) 30; dextrose (A.D.E.A Srl, Busto Arsizio, Italy) 30; malt extract (Costantino & C SpA, Favria, Italy) 15; $CaCO_3$ (Merck KGaA, Darmstadt, Germany) 4; and L-valine (Merck KGaA, Darmstadt, Germany) 1; pH of the medium prior to autoclaving was adjusted to 7.4 with NaOH. Cultivations in bioreactors were carried out at 30 °C, with stirring at 500 to 700 rpm (corresponding to 1.17 to 1.64 m/s of tip speed) and 2 L/min aeration rate. Foam production was controlled by the addition of Hodag antifoam through an antifoam sensor. A total of 25 mL of each culture was extracted every day for 1 week of fermentation [11,12,15].

### 2.2. A40926 Extraction and Analysis

A40926 was extracted by mixing 1 volume of mycelium and 3 volumes of borate buffer (100 mM $H_3BO_3$, 100 mM NaOH, pH 12; both reagents from Merck KGaA, Darmstadt, Germany). Samples were then centrifuged (16,000× *g* for 15 min) and incubated for 1 h at 50 °C. The glycopeptide-containing supernatant was filtered through a Durapore mem-

brane filter (0.45 μm) (Merck Millipore, Burlington, MA, USA). Glycopeptide production was estimated by High Performance Liquid Chromatography (HPLC) performed on a 5 μm-particle-size Ultrasphere ODS (Beckman Coulter Inc, Brea, CA, USA) column (4.6 by 250 mm) eluted at a flow rate of 1 mL/min with a 26-min linear gradient from 25% to 37% of phase B. Phase A was 20 mM HCOONH$_4$ (pH 4.5) (Merck KGaA, Darmstadt, Germany) -CH$_3$CN (Carlo Erba Reagents Srl, Cornaredo, Italy) (95:5 (*v/v*)), and Phase B was 20 mM HCOONH$_4$ (pH 4.5)-CH$_3$CN (5:95 (*v/v*)) mixture. Chromatography was performed with a VWR Hitachi diode array L-2455 HPLC system (VWR International, Radnor, PA, USA) with detection at 254 nm. A40926 purchased from Merck KGaA (Darmstadt, Germany) (purity of > 95%) was used as an internal standard [11,12]. A total of 10 mL of culture was collected from a parallel set of flasks for Packed Mycelium Volume (PMV %) and pH determination. Biomass production was also estimated as dry weight after 24 h incubation in an 80 °C oven and glucose consumption by using Diastix sticks (Bayer, Leverkusen, Germany).

### 2.3. Minimal Inhibitory Concentrations (MICs) and Population Analysis Profile (PAP)

MICs and PAP were determined as follows. Hyphae were sonicated as above described in order to disperse as much as possible the mycelial bacterial population and then seeded on Medium V0.1 agar (concentrations in g/L: soluble starch (Merck KGaA, Darmstadt, Germany) 2.4; glucose (A.D.E.A Srl, Busto Arsizio, Italy) 0.1; meat extract (Costantino & C SpA, Favria, Italy) 0.3; yeast extract (Costantino & C SpA, Favria, Italy) 0.5; tryptose (Merck KGaA, Darmstadt, Germany) 0.5; and agar (Merck KGaA, Darmstadt, Germany), 15; pH 7.2) [14] supplemented with different glycopeptide concentrations. Up to $10^7$ colony-forming units (cfu) were spread per each agar plate and incubated at 28 °C for 15 days. The MIC values were determined as the lowest antibiotic concentrations that inhibited visible growth after 10 days of incubation. The PAPs were determined as the surviving fraction of the population for each antibiotic concentration tested. The following A40926 ((Merck KGaA, Darmstadt, Germany)) concentrations were used: 0, 0.5, 1, 1.5, 2, 2.5, 3, 3.5, 4, 4.5, 5, 6, 7, and 8 μg/mL [11].

### 2.4. VanYn-Related Activity Measurement

D,D-carboxypeptidase activity was measured in VSP medium [14] at different time points from inoculum. Mycelial lysates were prepared as described previously [17]. The enzyme activity releasing D-Ala from the tripeptide *N*-acetyl-L-Lys-D-Ala-D-Ala (10 mM) was followed spectrophotometrically by a D-amino acid oxidase/peroxidase coupled reaction that oxidizes the colorimetric substrate 4-aminoantipyrine (Merck KGaA, Darmstadt, Germany) to chinonemine [18]. D,D-carboxypeptidase activity was normalized to dry biomass weight, as previously reported [13,17]. One unit is defined as the amount of enzyme that is able to convert one micromole of substrate in one minute.

## 3. Results

### 3.1. Heterogeneous A40926 Resistance Profile in Nonomuraea gerenzanensis Population

As previously reported [11–13], in mycelial actinomycetes, due to their complex life cycle, the standard method used in unicellular bacteria to determine MICs and to select resistant mutants is compromised by the formation of multicellular aggregates and by the coexistence of cells in different physiological states (e.g., vegetative mycelium, aerial mycelium, and spores). A sonication/filtration procedure was thus needed to prepare a more homogeneous population represented by viable unbranched hyphae with sizes ranging from 1 to 5 μm. Sonicated hyphae of *N. gerenzanensis* were then plated in Medium V0.1 in the presence of increasing concentrations of A40926. The strain morphology appeared uniform in Medium V0.1 agar plates without antibiotic addition (non-selective condition, Figure 1a) and in the presence of very low concentrations of A40926 (0.5, 1, and 1.5 μg/mL). MIC of A40926 under these experimental conditions was 4 μg/mL. Using the sub-inhibitory concentration of 2 μg/mL (so far identified as selective condition), two colony phenotypes were easily distinguished within the microbial population (Figure 1b).

The same phenotypes were also observed in plates containing 3 or 3.5 µg/mL A40926, but in those cases, the number of colonies per plate tended to be significantly reduced. In non-selective condition, colonies were generally regular, round shaped, orange pigmented, and 2–3 mm in diameter, with a typical irregular, convolute surface in the center (Figure 1a). In the selective condition, two different-sized colonies were detectable: in the so-called large colonies (G, from grand), the diameter size was 4-5-fold that of the small ones (named petit, P). Their morphology, both of G or P colonies, was different than in non-selective condition, being more pinkish, round shaped, and rather flat but with a protruding center.

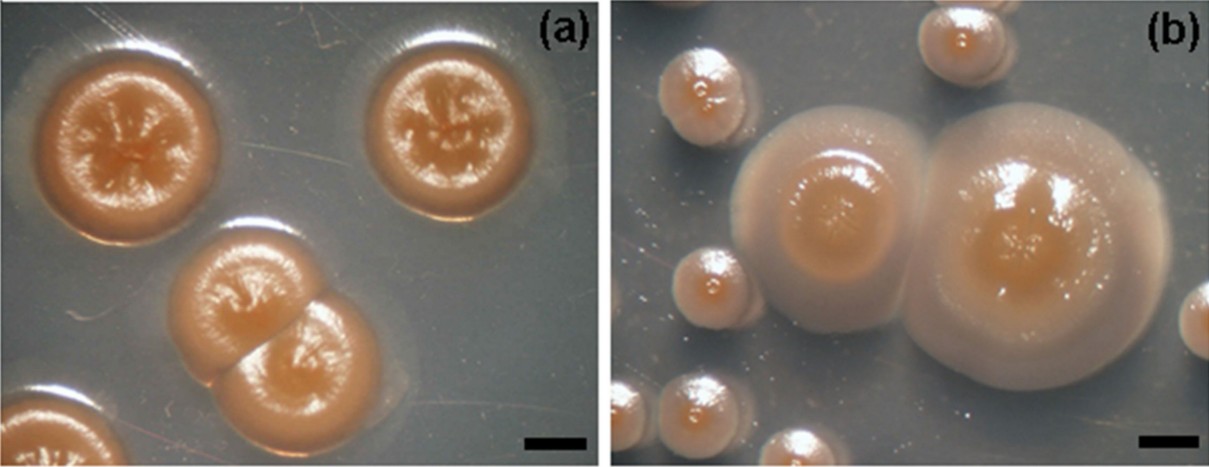

**Figure 1.** (**a**) Typical morphology of *N. gerenzanensis* colonies growing on Medium V0.1 agar without antibiotic addition: colonies appeared uniform in color, size, and shape. (**b**) Colonies growing in the presence of 2 µg/mL of A40926 showed a different morphology and dimensions: on the basis of their diameter, it was possible to distinguish P (petit, small) from G (grand, big) colonies. Bars are 0.8 mm.

Once isolated, these colonies were sub-cultivated in Medium V0.1 agar in the presence of 2 µg/mL of A40926. G phenotype appeared to be stable when sub-cultured in selective condition, whereas the P population, when sub-cultivated, continued to segregate G colonies with a frequency of 10 to 20% at each generation, indicating an intrinsic instability of this phenotype.

PAPs of the stable G population and of the wild-type indicated that A40926 MICs, which in these experimental conditions are considered the antibiotic concentrations that inhibits the growth of 99.9% of the population, were ca. 6 versus 3 µg/mL, respectively (Figure 2). In the unstable P population, the MIC was lower than in the wild-type (around 2.5 µg/mL) (Figure 2). These data confirm that *N. gerenzanensis* wild-type population has a heterogenous resistance profile that can probably vary according to the prevalence of a more or less resistant phenotype under the pressure of sub-inhibitory concentrations of A40926.

Since self-resistance in *N. gerenzanensis* was reported to be due to the action of a transmembrane D,D-carboxypeptidase encoded by the cluster-situated gene *dbv7*, which removes the last D-Alanine from the peptidoglycan precursors [17,18], we measured the D,D-carboxypeptidase activity in the membrane extracts of *N. gerenzanensis* wild-type and of its more resistant G population. The colorimetric-coupled assay detecting the amount of D-Ala released by the hydrolysis of *N*-acetyl-L-Lys-D-Ala-D-Ala tripeptide indicated that VanYn activity was two-fold higher in G cells than in the wild-type (respectively, 32 ± 2.5 vs. 15 ± 1.6 nmol/min*mg of total proteins, after 24 h from the induction by 2 µg/mL of A40926). These results confirm that VanYn enzyme activity correlates with the resistance phenotypes of G population and wild-type.

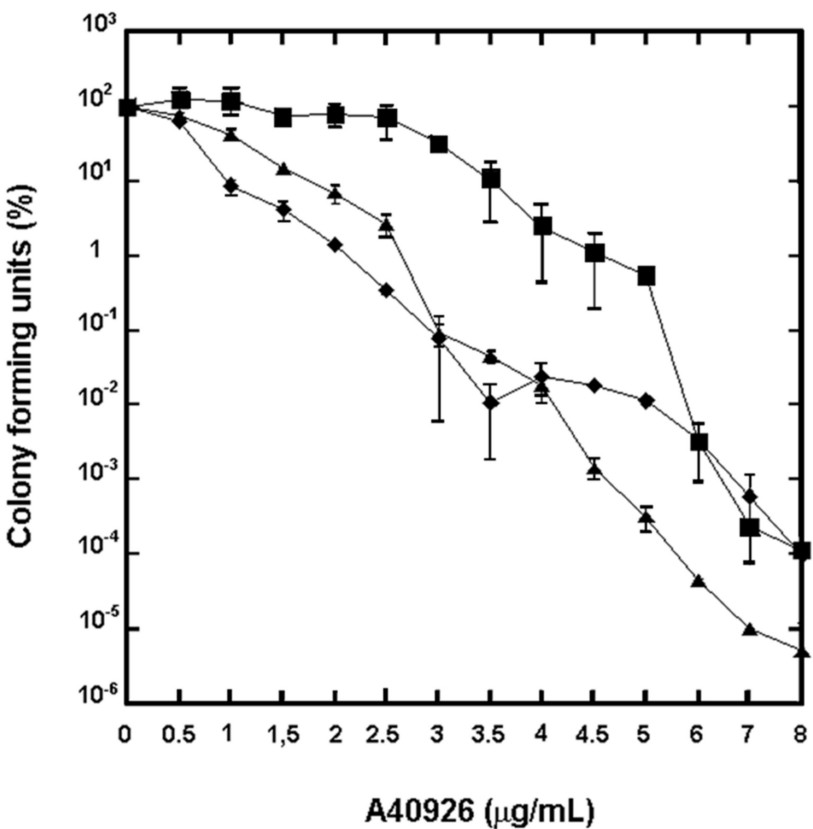

**Figure 2.** Population analysis profiles of *N. gerenzanensis*. Cfu growing in the presence of increasing concentrations of A40926 in the wild-type (filled triangles) and in the two selected populations, G (filled rectangles) and P (filled rhombi). Three colonies for each population were tested in parallel, and results are the average of three independent experiments. Bars indicate ± standard deviation.

*3.2. A40926 Production in N. gerenzanensis and in Its G and P Subpopulations*

Ten colonies for each population (G and P) were cultivated in parallel with the wild-type at flask level, and A40926 production was estimated by HPLC as reported in Materials and Methods. Results reported in Figure 3 indicate that the more resistant G colonies produce significantly more antibiotic than the wild-type (ca. 300 vs. 180 mg/L), whereas the less resistant P colonies produced slightly less (150 mg/L). This evidence corroborates the existence in the wild-type populations of a correlation between antibiotic A40926 production and self-resistance, which was previously demonstrated generating recombinant strains [12].

Upscaling the fermentation process at 2 L working-volume bioreactor scale, the growth curve and the production time course of the best producer G variant were compared with the wild-type profile (Figure 4a,b). In the G population, the dry biomass production was higher than in the wild-type, reaching the maximum value of 300 g/L after 120 h from inoculum versus the 200 g/L produced by the wild-type in the same cultivation time. Coherently, glucose consumption was faster in G population, being completely exhausted within the first 96 h of fermentation, whereas 144 h were needed for the wild-type to completely consume it. In the G population, A40926 production reached its maximum of 400 mg/L after 120 h from the inoculum versus the 150 mg/L produced in the wild-type after 144 h of cultivation. Thus, the G variant was confirmed at bioreactor scale to produce more than two-fold A40926 than the wild-type. It is worthy to note that the G variant conserved the same productivity in fermentation without the need to cultivate it in selective conditions. After twenty cycles of replication on non-selective conditions, no relevant change in morphology or antibiotic productivity was observed.

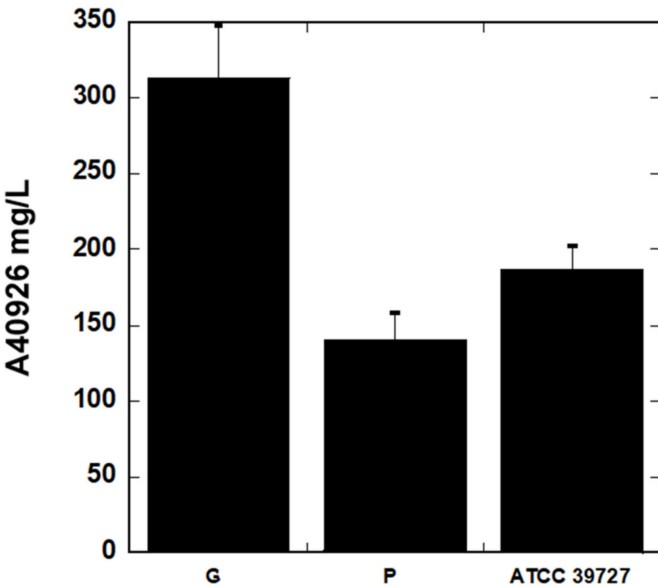

**Figure 3.** A40926 production, expressed in mg/L, of *N. gerenzanensis* ATCC 39727 and of its G and P variants grown in FM2 for 144 h after the inoculum in 500 mL Erlenmeyer flasks. Results given are the mean values from the cultivation and extraction of ten independent colonies per each condition. Bars indicate ± 1 standard deviation.

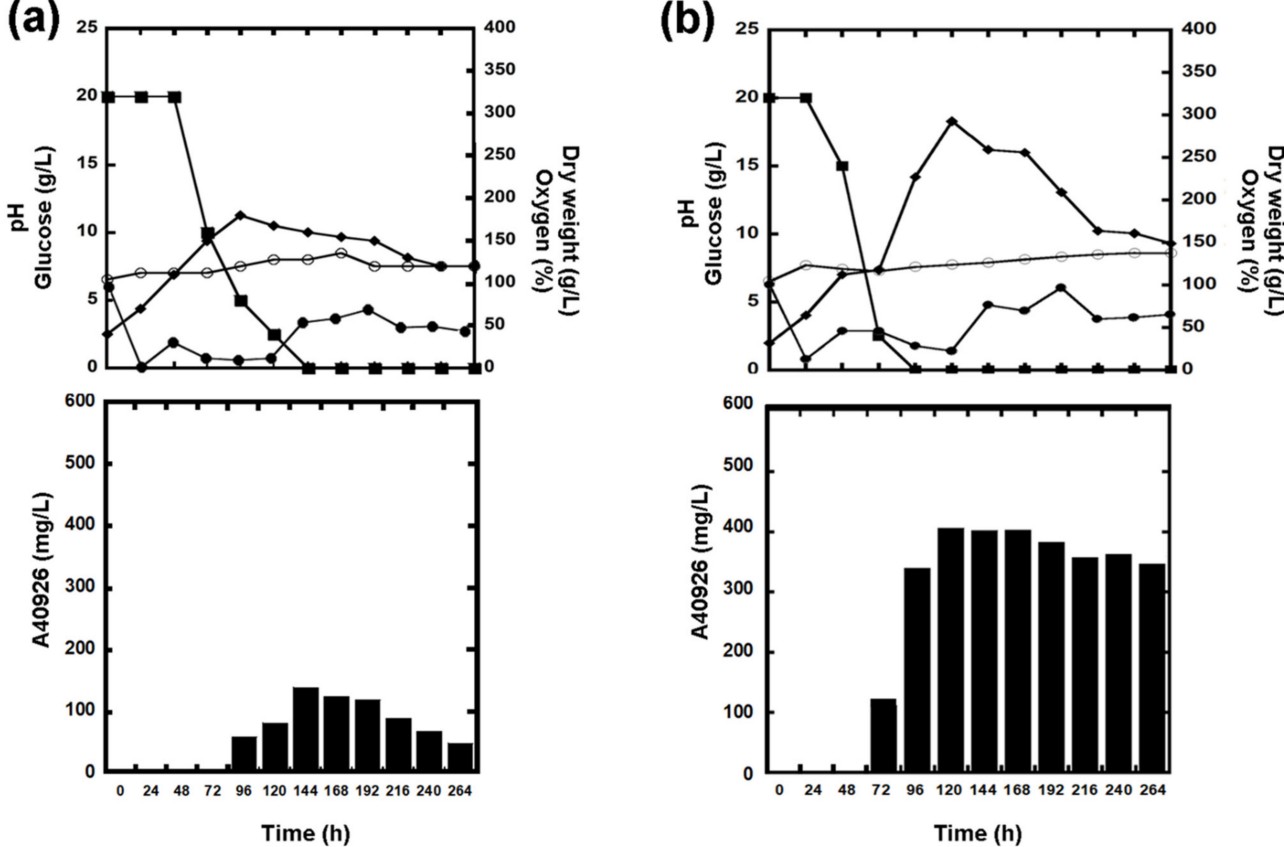

**Figure 4.** Time courses of *N. gerenzanensis* ATCC 39727 (**a**) and its more A40926-resistant G variant (**b**) cultivated in FM2 medium at 2 L working-volume bioreactor scale. Glucose consumption (filled rectangles), biomass accumulation (filled rhombi), pH (circles), $O_2$ level (filled circles), and A40926 production (filled bars) were monitored every 24 h.

## 4. Discussion

*Nonomuraea* is a genus of so-called rare or uncommon filamentous actinomycetes (to distinguish them from the most easy-to-isolate and -to-cultivate *Streptomyces* spp.), whose capability to produce specialized (secondary) metabolites is still rather poorly explored [19–21]. Probably, the most known *Nonomuraea* species is the producer of the antibiotic A40926, which was discovered in the late 1980s [1]. Notwithstanding its pharmaceutical relevance, the A40926 producer strain was properly classified at the species level three decades later [22]. At the same time, the complete annotation of its genome became available [23]. The final genome is ca. 12 Mb in size, organized in one main circular chromosome and three extra-chromosomal elements, and its main feature is the paralogous gene expansion, with many genes duplicated or expanded [23]. Another peculiar trait that was reported on *N. gerenzanensis* genome is that a large percentage of the genome (13.5%) is devoted to regulation, particularly transcription regulation. Additionally, a large number of genes appear involved in microbial adaptation under stressful conditions [23,24]. In this light, it is not surprising that we could distinguish different morphological phenotypes within the wild-type population when it was subjected to stress conditions. The large genome of this filamentous actinomycete has some degree of intrinsic instability/plasticity, and its versatile specialized (secondary) metabolism is under the control of sophisticated regulatory circuits that allow differential gene expression in response to varying environmental and cultivation conditions [23–25].

It is known that the chromosome of the mostly investigated *Streptomyces* spp. is very unstable, and it undergoes very large deletions spontaneously at rates higher than 0.1% of spores [25]. High-copy-number tandem amplifications of specific chromosomal regions are frequently associated with the deletions, and RecA seems to be involved in the amplification mechanism and in the control of genetic instability. It is likely that a similar molecular mechanism occurs in *Nonomuraea* spp. (so far, only three complete assemblies of *Nonomuraea* spp. genomes are available [20]), although this aspect has not been investigated yet. Further studies are thus needed to unveil the molecular mechanisms underlaying the genetic instability we observed in *N. gerenzanensis* wild-type population and in its P variant.

What is promising from the practical point of the strain improvement of *N. gerenzanensis* is that, under the pressure of the sub-inhibitory concentrations of the glycopeptide antibiotic that it produces, we could select morphologically diverse phenotypes to whom a different level of antibiotic production was associated. These results confirm the co-regulation of A40926 biosynthesis and self-resistance towards the produced glycopeptide, as suggested by the organization of the A40926 biosynthetic gene cluster, named *dbv* [5,11,12]. A40926 is a potent antibiotic active towards gram-positive bacteria, and its producing microorganism belongs to the gram-positive group. Consequently, *N. gerenzanensis* needs to protect itself during antibiotic production in order to avoid suicide [11–13]. In the light of the recent report that the real final biosynthesis product seems to be the acetyl-form of A40926 [9,10], it would had been interesting to test this last molecule, which unfortunately is not commercially available, in parallel with A40926. It is indeed known that A40926 and its acetyl-form are equipotent in antibacterial activity against most of the bacterial strains tested [1].

The only determinant of self-resistance in *N. gerenzanensis* so far reported is the expression of the *vanYn* gene (*dbv7*, clustered with the biosynthetic genes), encoding a D,D-carboxypeptidase, which converts the glycopeptide-sensitive peptidoglycan precursor of cell-wall biosynthesis into its resistant counterpart [11,17,18]. It was described [10] that the *dbv7* expression is induced by A40926, and it is repressed by its acetyl biosynthetic derivate in a feed-forward mechanism that reveals a complex regulatory circuit. Here, we reported that in the more abundantly growing and stable phenotype G, which produces nearly twice A40926 than its parental strain, VanYn enzyme activity was about the double than in the wild-type, suggesting that the increased self-resistance is associated with the improved antibiotic productivity, as recently highlighted in engineered recombinant

mutants [11,12]. Other authors previously reported that selecting for self-resistance leads to an improved productivity in antibiotic-producing actinomycetes [26]. Amplification of DNA segments, including antibiotic biosynthetic genes clustered with self-resistance genes, might represent a common molecular mechanism leading to an increased production in industrial strains selected in the presence of the produced antibiotics. For instance, in *Streptomyces kanamyceticus,* the level of kanamycin production depended on the copy number of its biosynthetic gene cluster, suggesting that DNA amplification occurred during strain improvement as a consequence of selection for increased kanamycin resistance [27].

In industrial programs of strain and fermentation improvement, the so-called strain maintenance protocol is routinely applied, based on the selection of the best clonal populations growing on agar media to then scale them up in large-scale, submerged, culture-based fermentations. Selection at this stage is often based on phenotypic traits (colony morphology, color, or simply increased productivity) that are basically applied blindly [28]. In the present work, we found a simple and rational way to discriminate high-producing colonies from *N. gerenzanensis* wild-type mixed population by using sub-inhibitory concentrations of the commercially available antibiotic A40926. A different colony morphology in plate was associated with the increased resistance and antibiotic productivity both at flask and at bioreactor scale, confirming that the empirical practice based on colony morphology observation still represents a valid tool, especially if applied to the original wild-type isolates, which exhibit some intrinsic heterogeneity. Although recombinant engineering is increasingly attracting industrial interest, allowing specific genetic modifications in the wild-type background, a preliminary analysis of the wild-type population is recommended to avoid misinterpretation of the results. In the case of *N. gerenzanensis,* we suggest that any further step of genetic manipulation should be applied to the stable selected G variant, which produces twice more than the wild-type. Finally, the fact that this higher-producing variant is more resistant to the produced glycopeptide antibiotic suggests that a possible way to further improve the A40926-production process could be the selection/construction of increasingly resistant A40926 mutants and/or the continuous removal of the antibiotic product during the fermentation process. Increasing the yield of the A40926 production might significantly contribute to the cost reduction of its clinically relevant semisynthetic derivative, dalbavancin.

## 5. Conclusions

Heterogeneous A40926 self-resistance profile in *N. gerenzanensis* population has permitted the isolation of two morphologically distinct populations to which a different pattern of A40926 production is associated. Under the experimental conditions used here, the more resistant variant produces more biomass and more antibiotic, reaching the maximum productivity of 400 mg/L A40926 at bioreactor scale. These results might contribute to developing a more sustainable process for producing A40926 and its semi-synthetic derivative, dalbavancin.

**Author Contributions:** Conceptualization, F.B. (Fabrizio Beltrametti) and F.M.; investigation, E.B., F.B. (Fabrizio Beltrametti), and A.B.; methodology, E.B., F.B. (Francesca Berini), and A.B.; supervision, F.M. and F.B. (Fabrizio Beltrametti); writing—original draft preparation, E.B. and F.B. (Fabrizio Beltrametti); writing—review and editing, F.M., E.B., F.B. (Fabrizio Beltrametti), A.B., and F.B. (Fabrizio Beltrametti). All authors have read and agreed to the published version of the manuscript.

**Funding:** This research was funded by FAR 2019–2020 (Fondi di Ateneo per la Ricerca, University of Insubria) to F.M.

**Informed Consent Statement:** Not applicable.

**Data Availability Statement:** Data is contained within the article.

**Acknowledgments:** The authors acknowledge the support of CIB (Consorzio Interuniversitario per le Biotecnologie) to F.M., E.B., and F.Ber.

**Conflicts of Interest:** The authors declare no conflict of interest.

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
