# Peer review of "Heterogeneous A40926 Self-Resistance Profile in Nonomuraea gerenzanensis Population Informs Strain Improvement"

_fermentation, doi:10.3390/fermentation7030140_

Round 1
Reviewer 1 Report
Binda et al. describe an innovative strategy to boost the production of the antibiotic precursor A40926 by the actinomycete N. gerenzanensis, essentially by selecting the largest colonies appearing on plates that contain sub-inhibitory concentrations of the A40926 molecule itself. The rationale behind this remarkably simple approach lies in the fact that A40926 resistance and biosynthesis tend to be strongly correlated, as was already suggested by earlier work that the authors cite.
The study is technically sound, well presented and clearly of considerable interest to anyone aiming to improve their A40926 production procedures. Moreover, as the authors duly note, the approach may be applicable not only to N. gerenzanensis but also to other antibiotic-producing microorganisms.
The only criticism I have is that (apart from the increase in VanYn activity that is reported) the manuscript provides very limited insight into the actual mechanisms behind the changes in A40926 production and resistance. Also, certain observations seem a bit puzzling, but are not discussed in any great detail. In particular, the authors propose (l. 137-140) that the changes they observe derive from (genetic?) variation/heterogeneity already present in the original population, but do not explain the fact that the sub-cultivated P form continues to segregate 10-20% G colonies per generation (l. 126). This percentage seems too high to arise from de novo mutations, so what mechanism do the authors believe might be responsible for the instability of the P phenotype?
The authors do mention that a large portion of the N. gerenzanensis genome is dedicated to gene regulation and in particular to transcription control (l. 195-200). Transcriptional de-repression of resistance genes in the presence of antibiotics is of course very common in bacteria, but I do not immediately see how (and why), in the case of N. gerenzanensis, this type of regulatory mechanism would lead to a stable sub-population of resistant cells in the absence of A40926. Do the authors have an explanation for this, or are they maybe aware of any other examples of this phenomenon in the literature?
No discussion is provided as to the gradual adaptation to A40926 that happens during fermentation (l. 38-39 and ref. 9). Do the authors believe this process to be comparable to the effects they observe on solid media? If so, what is the advantage of screening colonies on plates? It might be good to spell this out.
Finally, I was wondering about the technique of sonicating hyphae. The authors state that this serves "to reproduce as much as possible a unicellular bacteria situation" (l. 93), but do not provide any further details. Do the authors know what happens exactly when hyphae are sonicated? Could it be that this treatment somehow leads to an unequal distribution of components (proteins, nucleic acids, ...) to resulting daughter cells, which might readily explain some of the observations?
I fully understand that some of the questions above may not be simple to answer and possibly require additional work outside the immediate scope of the present manuscript. However, the authors may want to address these points in the discussion section, ideally in the light of any relevant literature.
Minor comments:
1. language errors and typos should be corrected, including the following:
- Hereby -> Here (l. 13 and l. 216)
- more than...that the wild type (l. 20)
- Notwithstanding -> Although (l. 33)
- biosyntehtic (l. 36)
- an heterogeneous (l. 42)
- cultural conditions (l. 48)
- a...phenotypes (l. 138-139)
- beetwen (l. 163)
- late 80' (l. 190)
- only (l. 191)
- fully annotation (l. 192)
- resulted of (l. 193)
- this...actinomycetes (l. 200-201)
- gifted (l. 201)
- diverse -> distinct (l. 245)
- hereby used -> used here (l. 246-247)
2. N. gerenzanensis is twice (l.26 and l. 187) qualified as a "rare" actinomycete, but no explanation is given as to what is meant by that and, in particular, why the word "rare" is in quotation marks.
Author Response
Dear Reviewer,
find below a point by point reply (in italics) to your questions/comments.
Sincerely
Fabrizio Beltrametti
Binda et al. describe an innovative strategy to boost the production of the antibiotic precursor A40926 by the actinomycete N. gerenzanensis, essentially by selecting the largest colonies appearing on plates that contain sub-inhibitory concentrations of the A40926 molecule itself. The rationale behind this remarkably simple approach lies in the fact that A40926 resistance and biosynthesis tend to be strongly correlated, as was already suggested by earlier work that the authors cite.
The study is technically sound, well presented and clearly of considerable interest to anyone aiming to improve their A40926 production procedures. Moreover, as the authors duly note, the approach may be applicable not only to N. gerenzanensis but also to other antibiotic-producing microorganisms.
We thank the Reviewer for having appreciated our work.
The only criticism I have is that (apart from the increase in VanYn activity that is reported) the manuscript provides very limited insight into the actual mechanisms behind the changes in A40926 production and resistance. Also, certain observations seem a bit puzzling, but are not discussed in any great detail. In particular, the authors propose (l. 137-140) that the changes they observe derive from (genetic?) variation/heterogeneity already present in the original population, but do not explain the fact that the sub-cultivated P form continues to segregate 10-20% G colonies per generation (l. 126). This percentage seems too high to arise from de novo mutations, so what mechanism do the authors believe might be responsible for the instability of the P phenotype?
We agree with the reviewer that in this paper we did not investigate in detail the molecular mechanism underlaying the genetic instability of P colonies. Our paper is indeed more focussed on the use and characterization of the selected G colonies to produce an increased amount of A40926. To this purpose, we agree with the last statement of the reviewer (see below) who comprehends that investigating this molecular mechanism maybe out of the immediate scope of this paper, but suggests us to address this point (together with the others raised below) in the discussion section, in the light of any relevant literature. We widely revised the discussion following the reviewer’ comments. We added in the discussion a comment on the genetic instability/plasticity (two sides of the same coin) of actinomycete chromosomes and we referred to what reported in the literature about the molecular mechanism of DNA amplification underlying the selection of high producing mutants resistant to their produced antibiotic. Three novel references were added (Volff, J.N.; Altenbuchner, J. Genetic instability of the Streptomyces chromosome. Mol Microbiol. 1998, 27, 239-46. doi: 10.1046/j.1365-2958.1998.00652.x; Yanai, K.; Murakami, T.; Bibb, M. Amplification of the entire kanamycin biosynthetic gene cluster during empirical strain improvement of Streptomyces kanamyceticus. Proc Natl Acad Sci U S A. 2006, 103, 9661-9666. doi: 10.1073/pnas.0603251103; Katz, L.; Baltz, R.H. Natural product discovery: past, present, and future. J. Ind. Microbiol. Biotechnol. 2016, 43, 155-176. doi: 10.1007/s10295-015-1723-5), and discussed.
Please see the revised tracked changed manuscript.
The authors do mention that a large portion of the N. gerenzanensis genome is dedicated to gene regulation and in particular to transcription control (l. 195-200). Transcriptional de-repression of resistance genes in the presence of antibiotics is of course very common in bacteria, but I do not immediately see how (and why), in the case of N. gerenzanensis, this type of regulatory mechanism would lead to a stable sub-population of resistant cells in the absence of A40926. Do the authors have an explanation for this, or are they maybe aware of any other examples of this phenomenon in the literature?
In the revised discussion, we reported about the possible mechanisms that could lead to a stable subpopulation of resistance cells in the absence of A40926. The clue difference between the antibiotic producers and the pathogenic bacteria (where indeed resistance determinants haves been mostly studied) is that self-resistance genes are clustered with the antibiotic biosynthetic genes and their expression is often co-regulated. This point has now better highlighted in the revised discussion. We also cited the paper reporting that in Streptomyces kanamyceticus, the level of kanamycin production depended on the copy number of its biosynthetic gene cluster, suggesting that DNA amplification occurred during strain improvement as a consequence of selection for increased kanamycin resistance (Yanai, K.; Murakami, T.; Bibb, M. Amplification of the entire kanamycin biosynthetic gene cluster during empirical strain improvement of Streptomyces kanamyceticus. Proc Natl Acad Sci U S A. 2006, 103, 9661-9666. doi: 10.1073/pnas.0603251103).
On the other hand, as recently reported by Alduina et al. (Alduina, R.; Tocchetti, A.; Costa, S.; Ferraro, C.; Cancemi, P.; Sosio, M.; Donadio, S. A two-component regulatory system with opposite effects on glycopeptide antibiotic biosynthesis and resistance. Sci. Rep. 2020, 10, 6200. DOI: 10.1038/s41598-020-63257-4) (this recent paper has now been cited in the revised manuscript), the regulatory circuit controlling self-resistance and glycopeptide production at transcriptional level in N. gerenzanensis is far to be fully understood and further investigations are needed to comprehend the role and interconnection of the four regulatory genes present in the dbv cluster. A comment in this sense has been introduced in the revised discussion.
No discussion is provided as to the gradual adaptation to A40926 that happens during fermentation (l. 38-39 and ref. 9). Do the authors believe this process to be comparable to the effects they observe on solid media? If so, what is the advantage of screening colonies on plates? It might be good to spell this out.
In our previous extensive work on glycopeptide self-resistance, we reported that A40926 induced an increase of self-resistance during fermentation. Recently, other authors (Alduina, R.; Tocchetti, A.; Costa, S.; Ferraro, C.; Cancemi, P.; Sosio, M.; Donadio, S. A two-component regulatory system with opposite effects on glycopeptide antibiotic biosynthesis and resistance. Sci. Rep. 2020, 10, 6200. DOI: 10.1038/s41598-020-63257-4) suggested a more complex regulatory network in which A40926 induces self-resistance, whereas its acetylated derivative represses it in a feed forward mechanism that we commented in the revised discussion. Said that, we do not believe that mycelia clumps growing in liquid cultures might be comparable to colonies growing in solid media. Aerated and agitated submerged cultures represent artificial environments developed to allow process scaling up. Solid cultures better resemble the natural environment where usually actinomycetes grow and differentiate. Consequently, the use of solid media is commonly accepted for controlling morphology and contamination and for selecting and characterizing novel phenotypes. We think that this concept is now clearer in the last part of the discussion.
Finally, I was wondering about the technique of sonicating hyphae. The authors state that this serves "to reproduce as much as possible a unicellular bacteria situation" (l. 93), but do not provide any further details. Do the authors know what happens exactly when hyphae are sonicated? Could it be that this treatment somehow leads to an unequal distribution of components (proteins, nucleic acids, ...) to resulting daughter cells, which might readily explain some of the observations?
In mycelial actinomycetes, due to their complex life cycle, the standard method used in unicellular bacteria to determine MICs and to select resistant mutants is compromised by the formation of multicellular aggregates and by the coexistence of cells in different physiological states (e.g., vegetative mycelium, aerial mycelium, and spores). Additionally, mycelium clump formation could hinder the identification of “recessive” phenotypes. We understand the point raised by the reviewer, thus we added more information on the sonication/filtration procedure and its control in the Material and Method section. We explained in the Result section that this step is needed. Since, as written in the Results section, the strain morphology (after the growth of sonicated hyphae) was uniform in Medium V0.1 agar plates in the absence of antibiotic addition, we consider that sonication process was not the cause of our further observations.
I fully understand that some of the questions above may not be simple to answer and possibly require additional work outside the immediate scope of the present manuscript. However, the authors may want to address these points in the discussion section, ideally in the light of any relevant literature.
We followed the reviewer suggestion and extensively revised the discussion, adding relevant references.
Minor comments:
- language errors and typos should be corrected, including the following:
- Hereby -> Here (l. 13 and l. 216)
- more than...that the wild type (l. 20)
- Notwithstanding -> Although (l. 33)
- biosyntehtic (l. 36)
- an heterogeneous (l. 42)
- cultural conditions (l. 48)
- a...phenotypes (l. 138-139)
- beetwen (l. 163)
- late 80' (l. 190)
- only (l. 191)
- fully annotation (l. 192)
- resulted of (l. 193)
- this...actinomycetes (l. 200-201)
- gifted (l. 201)
- diverse -> distinct (l. 245)
- hereby used -> used here (l. 246-247)
Thank you, the manuscript has been carefully checked.
- N. gerenzanensis is twice (l.26 and l. 187) qualified as a "rare" actinomycete, but no explanation is given as to what is meant by that and, in particular, why the word "rare" is in quotation marks.
We removed the word “rare” from the introduction and we better explained the origin of this definition at the beginning of the discussion, including a novel citation (Lazzarini, A.; Cavaletti, L.; Toppo, G.; Marinelli, F. Rare genera of actinomycetes as potential producers of new antibiotics. Antonie Van Leeuwenhoek 2001, 79, 399-405). Actually, another definition that could be used is “uncommon” intending that these actinomycetes are more difficult to isolate and to cultivate in comparison to the mostly isolated and studied streptomycetes.
Reviewer 2 Report
The manuscript “Heterogeneous A40926 self-resistance profile in Nonomuraea gerenzanensis population informs strain improvement” by Binda et al. is focused on the possibility to improve the production of a glycopeptide antibiotic (A40926) by the so-called “strain maintenance” protocol. The authors selected colonies of Nonomuraea gerenzanensis showing a different morphology after being cultured in the presence of sub-inhibitory concentrations of A40926 and obtained a different resistance profile. More resistant colonies were characterized by increased production in a fermentation medium.
Major concerns:
The authors should report about the stability of the two phenotypes, for example by indicating how many times the two strains were grown in the absence of sub-inhibitory concentration of the antibiotic before being tested by fermentation.
The experiment to get the two phenotypes is not detailed. For example, how many sub-inhibitory concentrations were used? It is not clear if they tested “increasing concentrations of A40926” L112 or “2 μg/mL was used (selective condition)” L115. If they tested more, it could be of interest to report the concentrations that caused the different morphology.
The authors could comment on their results reporting other experimental observations. Nonomuraea produces acetyl-A40926 and A40926. A40926 and acA40926 (considered the end product of the synthesis) had a different effect on the transcription of the resistance (dbv7) and the biosynthetic genes (dbv23, dbv24) of the dbv cluster; specifically, acA40926 had a repressive effect, while the A40926 had a positive effect, thus confirming their results. The authors should specify the difference of acA40926 and A40926 in the introduction, comment that they used only A40926 to evaluate resistance profile in the results and comment in the discussion about the similarity of results obtained with already published results.
Minor concerns regard typos and are listed below:
L42 which interestingly shows a heterogeneous
L49 ATCC
L62 autolysate
L60-64 65-69 are difficult to follow with the recipes of the media in the square brackets; the sentences could be ameliorated.
L81 HPLC should be written entirely the first time
L93 the sentence “in order to reproduce as much as possible a unicellular bacteria situation” could be modified in “in order to disperse as much as possible unicellular bacteria”
L96 the word cfu should be written entirely the first time. Then in figure 2, cfu should be consistently used.
L131 on the basis of
L151-154 (data not shown)
L176 consumption
L199 phenotypes
L203 environmental
L230 the empirical practice
Author Response
Dear Reviewer,
find below a point by point reply (in italics) to your comments
Sincerely
Fabrizio Beltrametti
Reviewer 2
The manuscript “Heterogeneous A40926 self-resistance profile in Nonomuraea gerenzanensis population informs strain improvement” by Binda et al. is focused on the possibility to improve the production of a glycopeptide antibiotic (A40926) by the so-called “strain maintenance” protocol. The authors selected colonies of Nonomuraea gerenzanensis showing a different morphology after being cultured in the presence of sub-inhibitory concentrations of A40926 and obtained a different resistance profile. More resistant colonies were characterized by increased production in a fermentation medium.
Major concerns:
The authors should report about the stability of the two phenotypes, for example by indicating how many times the two strains were grown in the absence of sub-inhibitory concentration of the antibiotic before being tested by fermentation.
The G phenotype is currently used in our laboratories for A40926 production. The strain underwent at least 20 cycles of replication on non-selective conditions and displayed no change in morphology or titre. Thus, information has been added in the revised manuscript, at the end of the result section. The P phenotype was not considered further in our studies due to instability and to low productivity.
The experiment to get the two phenotypes is not detailed. For example, how many sub-inhibitory concentrations were used? It is not clear if they tested “increasing concentrations of A40926” L112 or “2 μg/mL was used (selective condition)” L115. If they tested more, it could be of interest to report the concentrations that caused the different morphology.
The experiments were performed with all the A40926 concentrations described in the PAP figure. The concentrations have been now reported in the Material and Methods section. Higher concentrations than 2 µg/mL caused the different morphology but the number of the colonies per plate was drastically reduced. This is coherent with the fact that the MIC was determined as 4 µg/mL.
The authors could comment on their results reporting other experimental observations. Nonomuraea produces acetyl-A40926 and A40926. A40926 and acA40926 (considered the end product of the synthesis) had a different effect on the transcription of the resistance (dbv7) and the biosynthetic genes (dbv23, dbv24) of the dbv cluster; specifically, acA40926 had a repressive effect, while the A40926 had a positive effect, thus confirming their results. The authors should specify the difference of acA40926 and A40926 in the introduction, comment that they used only A40926 to evaluate resistance profile in the results and comment in the discussion about the similarity of results obtained with already published results.
We agree with the reviewer and we explained the difference of acA40926 and A40926 in the introduction by also adding additional references (Alt, S.; Bernasconi, A.; Sosio, M.; Brunati, C.; Donadio, S.; Maffioli, S. I. Toward single-peak dalbavancin analogs through bi-ology and chemistry. ACS Chem. Biol., 2019 14, 356−360. doi: 10.1021/acschembio.9b00050. Alduina, R.; Tocchetti, A.; Costa, S.; Ferraro, C.; Cancemi, P.; Sosio, M.; Donadio, S. A two-component regulatory system with opposite effects on glycopeptide antibiotic biosynthesis and resistance. Sci. Rep. 2020, 10, 6200. DOI: 10.1038/s41598-020-63257-4). We used only A40926 to evaluate resistance profile in the results since this molecule is the commercially available antibiotic. Acetyl A40926 is not available. We commented in the discussion about the similarity of our results obtained with those already published, citing the recent work of Alduina et al. (Alduina, R.; Tocchetti, A.; Costa, S.; Ferraro, C.; Cancemi, P.; Sosio, M.; Donadio, S. A two-component regulatory system with opposite effects on glycopeptide antibiotic biosynthesis and resistance. Sci. Rep. 2020, 10, 6200. DOI: 10.1038/s41598-020-63257-4).
Minor concerns regard typos and are listed below:
L42 which interestingly shows a heterogeneous
L49 ATCC
L62 autolysate
L60-64 65-69 are difficult to follow with the recipes of the media in the square brackets; the sentences could be ameliorated.
L81 HPLC should be written entirely the first time
L93 the sentence “in order to reproduce as much as possible a unicellular bacteria situation” could be modified in “in order to disperse as much as possible unicellular bacteria”
L96 the word cfu should be written entirely the first time. Then in figure 2, cfu should be consistently used.
L131 on the basis of
L151-154 (data not shown) VanY activity data are shown in brackets
L176 consumption
L199 phenotypes
L203 environmental
L230 the empirical practice
Thank you, the manuscript has been carefully checked.
Round 2
Reviewer 2 Report
The authors addressed all my concerns and improved the manuscript, which I consider suitable for publication in Fermentation.
regards